# Measurement of Equivalent BRDF on the Surface of Solar Panel with Periodic Structure

**Qingyu Hou, Zhile Wang, Jinyu Su and Fanjiao Tan \***

Research Center for Space Optical Engineering, Harbin Institute of Technology, Harbin 150001, China; houqingyu@126.com (Q.H.); wangzhile@hit.edu.cn (Z.W.); loveu.myprettygirl@foxmail.com (J.S.)
**\*** Correspondence: tanfanjiao@hit.edu.cn

**Abstract:** The surface of a complex material with a periodic structure is equivalent to the surface of a uniform material, and the measurement and data processing methods for the equivalent optical BRDF of a solar panel based on a large-scale spot are proposed. Based on a solar simulator, high-intensity light illuminometer, low-intensity light illuminometer and precise rotary platform, the BRDF automatic measurement platform is built. Spot size optimization and a measurement radius optimization method are proposed. The measurement results show that the BRDF of the solar panel has specular reflection characteristics, but that it still differs from the solar cell in the half width of the BRDF curve. Measurement error analysis is performed for the measurement process; its value is 6.74%. The measurement results can be used to improve the understanding of the optical reflection characteristics of the solar panel. Meanwhile, the method can also be used to measure and characterize the coatings of heat insulation material and scattering coating. The measurement data also has practical reference value for evaluations of improvements of the light absorption of the surface functional material. Finally, it can be used to simulate the target image scene.

**Keywords:** measurement; equivalent BRDF; solar panel; error analysis

## 1. Introduction

The bidirectional reflectance distribution function (BRDF) is gradually introduced to describe the reflection characteristics of the material surfaces; it is widely used in the target scattering characteristics description [1], radiation calibration [2] and remote sensing imaging [3]. Measurement of optical BRDF on the surface of the space target has great value in the modeling of space target scattering characteristics [4], spectral feature extraction [5] and target identification and tracking.

Researchers have done a lot of work on the optical BRDF measurements of space target materials. Wang [6] in 2009, Wang [7], Yuan [5] in 2010, Wang [8] in 2013 and Hou [9] in 2014 built BRDF measurement platforms and gradually improved the scheme from the perspective of multi-spectral automatic measurement; they then carried out BRDF measurements of space target materials, such as yellow thermal control material, silver thermal control material and solar cells [10]. At the same time, in terms of modelling and simulating the on-orbit imaging characteristics of space targets, the material surface was divided into facet elements, and the photometric signals of facet elements based on BRDF were projected onto the image plane. Then, the characteristics of the imaging system were combined to complete the imaging characteristics simulation of the space target. The simulation results have certain reference values in the optical characteristics of the space target [11,12].

In the above research, the BRDF of the facet element of uniform material surfaces is usually measured. However, in practical applications, the surface of the solar panel has support structures in addition to the solar cell. Solar panels comprise a complex material surface with periodic structures, and there are some differences in the optical reflection characteristics of solar panels and solar cells.

Therefore, the accuracy of space target imaging characteristics simulation based on the measured BRDF of the solar cell cannot be satisfied.

In order to solve this problem, solar panels with a complex structure surface are equal to a uniform material surface; as such, optical reflection characteristic measurement and a data processing method are proposed in this paper. An optical BRDF automatic measurement platform for complex material surfaces is established, and the high-precision measurement of equivalent optical BRDF on solar panel is achieved. The measurement results can be used to improve the understanding of the optical reflection characteristics of solar panel. Meanwhile, the equivalent optical BRDF can be used for imaging simulation, which can improve the precision of the simulation [13–17].

## 2. Equivalent BRDF of Complex Material Surface

### 2.1. BRDF of Uniform Material Surface

The bidirectional reflectance distribution function (BRDF) an inherent physical property of the material surface, and describes the light reflective characteristic in each direction on the hemisphere when the light source is incident from a certain direction [5–9]. The geometric relationship of the bidirectional reflectance distribution function is shown in Figure 1.

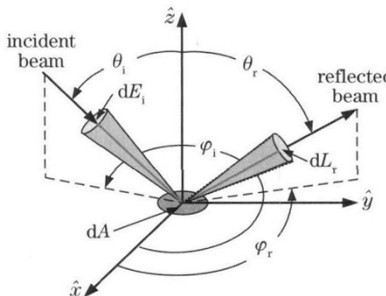

**Figure 1.** Incidence and reflected vectors of BRDF model [6].

Bidirectional reflectance distribution function is the ratio of the reflected radiance and the incident irradiance, the mathematical expression for which is [18]:

$$f_r(\theta_i, \varphi_i; \theta_r, \varphi_r; \lambda) = \frac{dL_r(\theta_i, \varphi_i; \theta_r, \varphi_r; \lambda)}{dE_i(\theta_i, \varphi_i; \lambda)} \tag{1}$$

where $\theta_i$, $\varphi_i$ are the incident zenith and incident azimuth angles respectively, $\theta_r$, $\varphi_r$ are the reflected zenith and reflected azimuth angles respectively, $dL_r$ is the reflection radiance of surface facet $dA$ in the $\theta_r$, $\varphi_r$ direction, $dE_i$ is the incident irradiance of surface facet $dA$ in the $\theta_i$, $\varphi_i$ direction.

### 2.2. Definition of Equivalent BRDF

Solar panel surfaces are structures composed of silicon wafers and special metals, as shown in Figure 2. The cross-section of the solar panel is shown in Figure 3, including the cover glass, ethylene-vinyl acetate (EVA), silicon solar cells with electrode and back glass [19]. The materials and structural forms of other types of solar panels may differ from this kind solar panel, but they all comprise complex material surfaces with periodic structures. When light reaches the junction between the silicon wafers or reaches different silicon wafers, the scattering characteristics are so different that the single silicon wafer's scattering characteristics cannot reflect the scattering characteristics of the whole solar panel. Therefore, it is necessary to extend the concept of BRDF and to measure the

scattering characteristics of a larger area in complex material. As such, we measured the intensity or illuminance (converted into intensity) at different scattering angles to obtain the function:

$$F(\theta_i, \varphi_i; \theta_r, \varphi_r) = \frac{L(\theta_r, \varphi_r)}{E(\theta_i, \varphi_i)} \tag{2}$$

Using Equation (2) to describe the scattering characteristics of solar panel, the solar panel surface can be the equivalent of a uniform material surface.

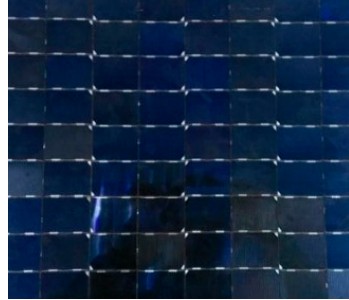

**Figure 2.** The surface structure of the solar panel.

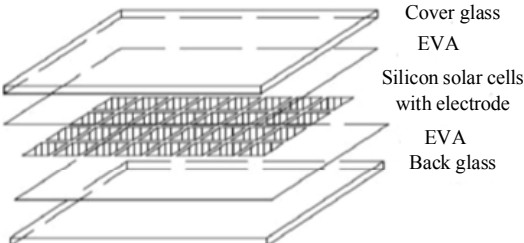

**Figure 3.** The cross-section of the solar panel [19].

## 3. Measuring Principle and Precision Improvement Strategy

This section may be divided by subheadings. It should provide a concise description of the experimental results and their interpretation, as well as the experimental conclusions.

### 3.1. Measuring Principle

As shown in Figure 4, the solar simulator emits a vertical downward light through a mirror placed in the degree of 45, and then reflects horizontally to the test sample.

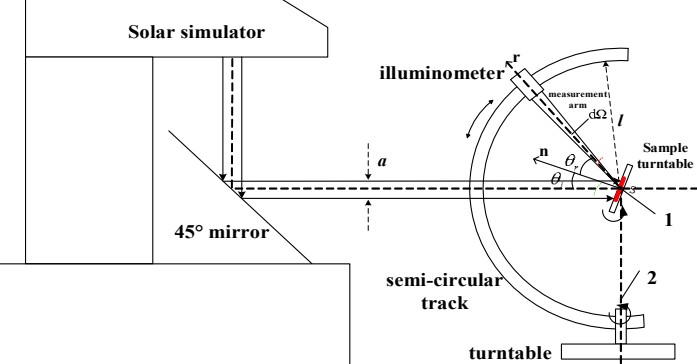

**Figure 4.** The measurement device of BRDF on the Surface of the Solar Panel.

The sample can rotate along the axis in the paper plane (axis 1). The angle between the normal vector **n** of the sample and the light incident direction (horizontal) determines the incident angle $\theta_i$.

The detector (illuminometer) was placed on the semi-circular track, and the detector could slide along the semi-circular track. The detector always faces the center of the sphere, so the detector received the scattered signals throughout the spherical space. The position of the detector on the semi-circular track and the rotation angle of the turntable (axis 2) determined the zenith angle $\theta_r$ and the azimuth angle $\varphi_r$, and determine the scattering direction **r**.

The detector was an illuminometer, the measured data from which was $E(\theta_r, \varphi_r)$. According to Equation (2), we need to transform $E(\theta_r, \varphi_r)$ to $L(\theta_r, \varphi_r)$. The luminous flux reflected from the test sample in solid angle $d\Omega$ was:

$$d\phi = L(\theta_r, \varphi_r) \cdot d\Omega \cdot \cos\theta_r \cdot S \tag{3}$$

where $S$ is the spot area on solar panel, $d\Omega$ is the solid angle and $d\Omega = A_0/l^2$, $l$ is the radius of semi-circular track.

The radiant flux receives by the illuminometer is:

$$d\phi' = E(\theta_r, \varphi_r) \cdot A_0 \tag{4}$$

The luminous flux that detector receives equals to the radiant flux of the test sample, it means $d\phi = d\phi'$. Then

$$L(\theta_r, \varphi_r) \cdot d\Omega \cdot \cos\theta_r \cdot S = E(\theta_r, \varphi_r) \cdot A_0 \tag{5}$$

Therefore

$$L(\theta_r, \varphi_r) = \frac{E(\theta_r, \varphi_r)l^2}{\cos\theta_r \cdot S} \tag{6}$$

where $S = \frac{S_0}{\cos\theta_i}$, and $S_0$ is spot area on solar panel when $\theta_i = 0$, then Equation (6) can be written as:

$$L(\theta_r, \varphi_r) = \frac{E(\theta_r, \varphi_r)\cos\theta_i l^2}{\cos\theta_r S_0} \tag{7}$$

Corresponding to different incident angles, the incident illumination on the sample is:

$$E_i = E_0 \cos\theta_i \tag{8}$$

where $E_0$ is irradiance when the incident light is vertical to the solar panel.

Then

$$F = \frac{L(\theta_r, \varphi_r)}{E_i} = \frac{E(\theta_r, \varphi_r)l^2}{\cos\theta_r S_0 E_0} \tag{9}$$

Since the solar panel was placed vertically, the step angle $\alpha$ of the detector on the semi-circular track and the rotation angle of the turntable were not the same as the zenith and azimuth angles in the BRDF model. Therefore, according to Equations (10) and (11), each zenith and azimuth angle needs to be converted into the step angle $\alpha$ of the detector on the semi-circular track and the rotation angle $\beta$ of the turntable, and the corresponding step angle $\alpha$ and rotation angle $\beta$ need to be adjusted to achieve the scattered illuminance acquisition in the hemisphere space. The conversion formula between $\theta_r$, $\varphi_r$ and $\alpha$, $\beta$ is:

$$\begin{cases} \cos\theta_r = \sin\beta \cdot \cos\alpha \\ \tan\varphi_r = \tan\alpha / \cos\beta \end{cases} \tag{10}$$

$$\begin{cases} \cos\alpha = \sqrt{\dfrac{1+\tan^2\varphi_r \cdot \cos^2\theta_r}{1+\tan^2\varphi_r}} \\ \sin\beta = \dfrac{\cos\theta_r}{\cos\alpha} = \sqrt{\dfrac{\cos^2\theta_r(\tan^2\varphi_r+1)}{1+\tan^2\varphi_r\cos^2\theta_r}} \end{cases} \tag{11}$$

### 3.2. Measurement Accuracy Improvement Strategy

#### 3.2.1. Spot Size Optimization

The material surface BRDF reflects the optical reflection characteristics of the material, and should be measured with a tiny light beam. However, the measurement of a tiny beam cannot reflect the overall scattering characteristic because the surface of solar panels contains a complex periodic structure. It is necessary to increase the beam size so that the spot can cover surface structure as much as possible; a minimum spot size should ensure that the measurement data of reflectance illumination do not change significantly with the translation of the solar panel.

#### 3.2.2. Measurement Radius Optimization

When the measurement radius $l$ is constant, if the spot size $S$ is too large, Equation (2) can be expressed as the sum of the reflection fluxes of all the facet elements, and can be written as:

$$\mathrm{d}\phi = \sum_{i=1}^{n} L^i(\theta_r^i, \varphi_r^i)\mathrm{d}\Omega^i \cos\theta_r^i s^i \tag{12}$$

where $S = \sum_{i=1}^{n} s^i$, $s^i$ is the area of the facet elements $i$; $\mathrm{d}\Omega^i$, $\theta_r^i$, $L^i(\theta_r^i, \varphi_r^i)$ are solid angle, zenith angle and radiance, respectively. Obviously, the larger the spot size $S$, the larger the range of $\theta_r^i$ corresponding to every $s^i$. The error of Equation (2), which is based on the whole spot size $S$ and the single zenith angle $\theta_r$, becomes larger, as does the calculation error of radiance $L(\theta_r, \varphi_r)$ and measurement error of BRDF. Therefore, in order to ensure the accuracy of $L(\theta_r, \varphi_r)$, the range of $\theta_r^i$ should be as small as possible. The way to solve this problem is to increase the radius $l$ of the semi-circular track (see Figure 5).

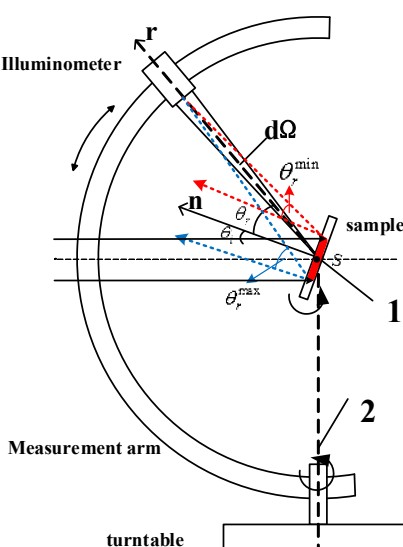

**Figure 5.** The right side of measurement device of solar panel.

Here, the range of $\theta_r^i$, which is defined by maximum angle deviation (MAD), can be expressed as:

$$MAD(\theta_r, \theta_i, S) = \max_{i}\left|\theta_r^i - \theta_r\right| \tag{13}$$

The limitation is that the maximum angle deviation of $\theta_r^i$ is less than 3° when $\theta_i = 0°$, $\theta_r \leq 60°$, so the measurement radius $l$ and the spot diameter $a$ must satisfy $l : a \geq 10 : 1$ when vertically irradiated.

Therefore, $l : a = 10 : 1$ is set as measurement parameter. When $\theta_i = 0°$, the relation between $\theta_r$ and the range of $\theta_r^i$ is shown in Table 1, the MAD of $\theta_r^i$ is 2.86°.

**Table 1.** The relation between $\theta_r$ and the range of $\theta_r^i$.

| Number | $\theta_r$ | Range of $\theta_r^i$ | MAD |
|:---:|:---:|:---:|:---:|
| 1 | 0° | 0°~2.86° | 2.86° |
| 2 | 30° | 27.47°~32.43° | 2.53° |
| 3 | 60° | 58.50°~61.37° | 1.50° |

## 4. Measurement Equipment and Measurement Process

### 4.1. Measurement Equipment

In the measurement structure, the ad-hoc test sample holder, semi-circular track and brackets were manufactured and constructed. The three-dimensional schematic diagram of the frame structure is shown in Figure 6.

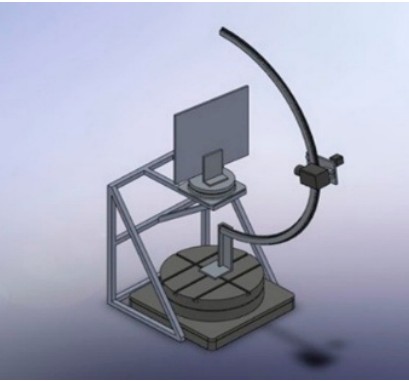

**Figure 6.** The three-dimensional schematic diagram of the frame structure.

The measurement equipment and performance parameters selected during the measurement are shown in Table 2. The selection of the detector has a great influence on the accuracy and the range of the measurement. Regarding measurements of the illuminance of the incident light from the solar simulator, detectors with large range were needed, so a high-intensity light illuminometer was selected. Scattering measurements in all directions include weak-light detection, so a low-intensity light illuminometer with high precision was selected.

**Table 2.** Technical data of the measurement devices.

| Name | Type | Parameter and Specification |
|:---:|:---:|:---|
| Solar simulator | 94123A | Collimated Angle: $< 90°$<br>The power of xenon lamp: 1600 W<br>Irradiated area:300 mm $\times$ 300 mm |
| High-intensity light illuminometer | TES1339 | Measurement range: 0.01~999900 lx<br>Measurement error: $\leq 3\%$ |
| Low-intensity light illuminometer | PHOTO-2000m | Measurement range: $10^{-6}$ ~1.0 lx<br>Measurement error: $\leq 4\%$ |
| Precise rotary platform | MRS102 | Angle Resolution: 0.00022°<br>Displacement Precision: <10 s of arc |

### 4.2. Measurement Process

**Step 1: The Horizontal Adjustment and Concentric Adjustment of the Measuring Device.**

The horizontal adjustment includes the adjustment of the sample turntable and the semi-circular track turntable, adjustment principle is shown in the Figure 7. The intersection of the incident light and the sample is the central location of the entire test environment; all equipment centers must be based

on this point. The detector was set to the middle of the semi-circular track, the supporting structure was adjusted to make the detector in a horizontal state, then the height of the detector was increased or reduced to make the detector signal up to maximum. Now the optical axis of the detector is coaxial with the input optical axis, and the detector points to the center of the semi-circular track.

The concentric adjustment principle is shown in the Figure 8. The 45° reflector was placed in the sample turntable, the reflective surface of the reflector was passed through the shaft of the sample turntable, and the semi-circular track was rotated 90° to receive the energy reflected from the reflector. The center position of the sample turntable was adjusted to maximize the detector reading. Then, the center of the semi-circular track was concentric with the axis of the sample turntable. Therefore, the center of the sample turntable and the semi-circular track turntable was adjusted to the same vertical line, which ensured that the direction of the detector always pointed to the center of the semi-circular track.

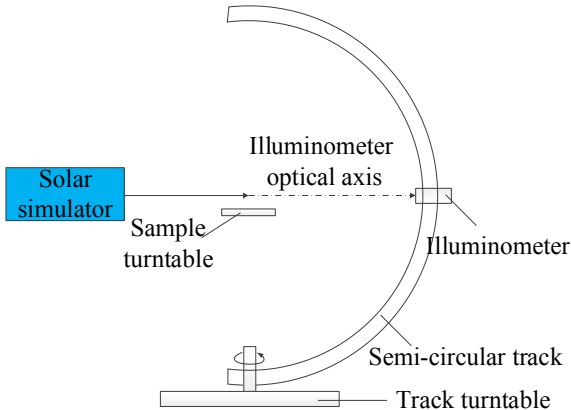

**Figure 7.** The horizontal adjustment sketch.

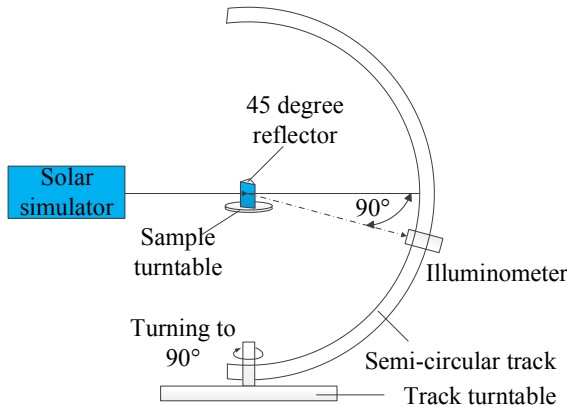

**Figure 8.** The concentric adjustment sketch.

**Step 2: Measurement Angle Calibration.**

The measurement angle calibration includes calibrating the initial position of the detector and the sample turntable.

The calibration principle of initial position of the detector is shown in the Figure 9. Firstly, the step motor of the track was controlled to bring the detector into the horizontal state; then, the step motor of the turntable rotated the semi-circular track to make the illuminometer's optical pupil point to the direction of the incident light, and the position of the detector was slightly adjusted to make the output signal maximum. In this case, the incident light directly entered the detector, which was in contrast to the measuring center. The position of the detector in semi-circular track and the angle of the turntable were recorded.

The calibration principle of initial position of the sample turntable is shown in the Figure 10. While the sample plane is vertical and facing the incident direction, it is regarded as the initial position of the sample plane. The sample plane was placed in the center of the turntable, and a small mirror was attached to the sample. Next, the stepper motor was manipulated to make the reflected spot coincident with the incident pupil completely. In this case, the position of the turntable was the initial position of the sample plane. Then, the initial incident zenith angle and initial incident azimuth angle were 0° and 0°, respectively.

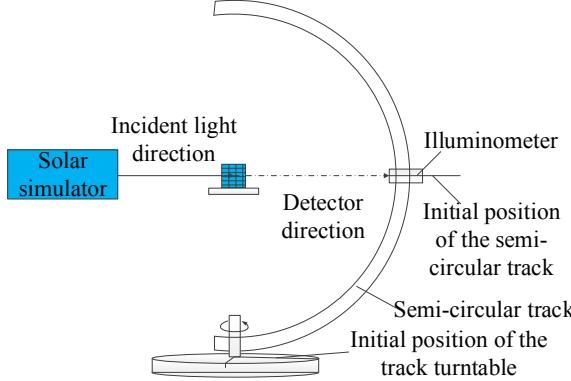

**Figure 9.** The sketch of calibration of initial position of the detector.

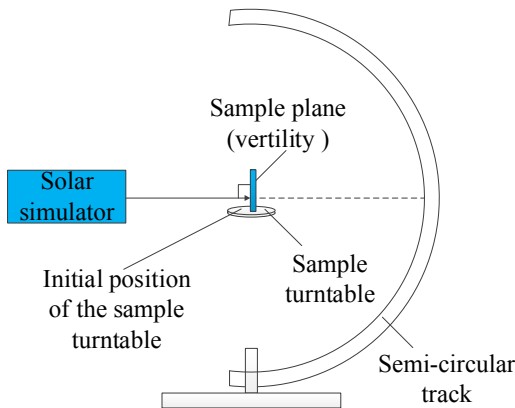

**Figure 10.** The sketch of calibration of initial position of the sample turntable.

**Step 3: Measuring the Incident Illuminance.**

About 30 min after starting up the sun simulator, it will work stably. The sample may then be removed, and it is possible to measure the average illuminance of the incident position and spot area.

**Step 4: Eliminating the Background Data.**

In order to improve the accuracy of the measurements, the background light must be measured, which includes the scattering of light source in dark room and ambient light in test space. Before the test, the light source was turned on and the solar panel was placed in the measuring position. The detector was then moved to several measuring positions in the hemispheric space. By comparison, it can be seen that the background light values are basically the same at these positions, so the influence of the background light can be regarded as a constant, and as such, are globally eliminated from the measured data. In this experiment, the background light is 0.02lx.

**Step 5: Measuring the Reflected Illuminance.**

The measurement process served to collect the detector signal at different incident angles and reflected angles. Firstly, the incident and reflection angles were adjusted; finally, the reflection radiance was collected.

The reflection angle includes the reflected zenith angle $\theta_r$ and the reflected azimuth angle $\varphi_r$, which can be obtained from the step angle $\alpha$ of the detector motor and the rotation angle $\beta$ of the semi-circular track turntable. For each detector signal measurement, the intervals of $\alpha$ and $\beta$ are 5° and 15°, respectively. According to Equations (10) and (11) in the original paper, each $\alpha$ and $\beta$ must be converted into zenith angle $\theta_r$ and azimuth angle $\varphi_r$.

**Step 6: Measurement Data Processing.**

Based on the above measurement parameters, the BRDF calculation is realized by Formula (9).

## 5. Measurement Results and Analysis

### 5.1. Spot Size Optimization Results

In this measurement experiment, we designed a light-shielding tube that was able to adjust the area of the light spot by changing the diameter of the tube. Since the size of the solar cell was 30 mm × 40 mm, the optional spot diameters were 120 mm, 170 mm and 240 mm, respectively. Under these three conditions, the measured BRDF values do not change with solar panel translating. Therefore, in measuring radius $l$ is 1.2 m according to $l : a = 10 : 1$, the BRDF calculating data in the case of three kinds of spots is shown in Table 3. It can be seen that, compared with the 120 mm spot, the larger the spot, the larger the BRDF error. The above conclusions are consistent with the analysis in Section 3.2.2.

**Table 3.** The BRDF calculating data of the three conditions of spot diameter.

|  | Diameter: 240 mm | Diameter: 170 mm | Diameter: 120 mm | Diameter: 120 mm (Translate Half Silicon Wafer) |
|---|---|---|---|---|
| Vertical projection area ($m^2$) | 0.045 | 0.023 | 0.011 | 0.011 |
| Incident illuminance (lx) | 25,500 | 14,500 | 6500 | 6500 |
| $\theta_i = 30°, \theta_d = 30°$ | 0.235019 | 0.222273 | 0.209133 | 0.208671 |
| $\theta_i = 30°, \theta_d = 60°$ | 0.052507 | 0.040240 | 0.033764 | 0.034827 |
| $\theta_i = 45°, \theta_d = 15°$ | 0.022246 | 0.019436 | 0.009998 | 0.009633 |
| $\theta_i = 45°, \theta_d = 45°$ | 0.464673 | 0.401097 | 0.374244 | 0.371373 |
| $\theta_i = 60°, \theta_d = 0°$ | 0.005353 | 0.005210 | 0.000872 | 0.000839 |
| $\theta_i = 60°, \theta_d = 60°$ | 1.325325 | 1.258932 | 1.145356 | 1.148140 |

### 5.2. Measurement Results

#### 5.2.1. Measurement Results of Solar Panel

The scattering distribution characteristics on the surface of the solar panels were measured at incident angles of 15°, 30°, 45° and 60°, respectively. The range of scattering zenith angle is 0~90° when the angle step was 5°, and was −180°~180° when the angle step was 5°. The measurement result is shown in Figure 11. Obviously, the scattering angle corresponding to the maximum value is generally in the specular reflection direction.

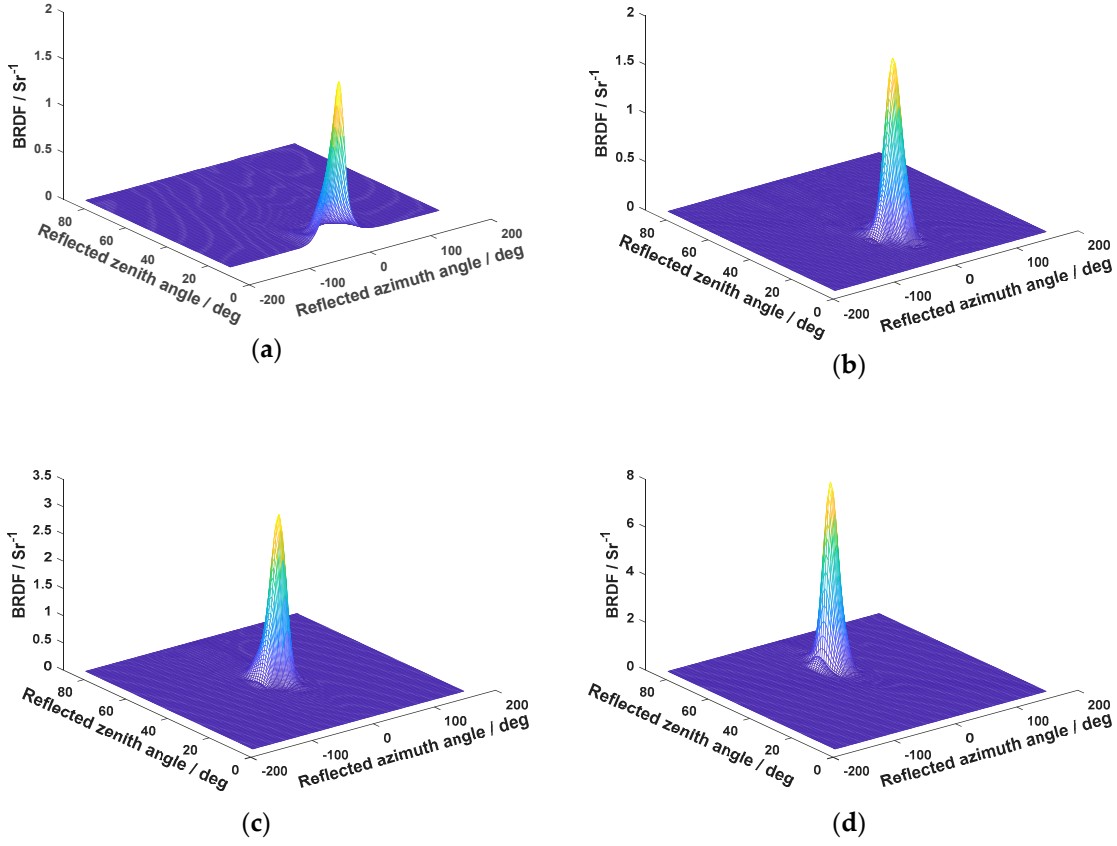

**Figure 11.** BRDF distribution of solar panel. (**a**) $\theta_i = 15°$; (**b**) $\theta_i = 30°$; (**c**) $\theta_i = 45°$; (**d**) $\theta_i = 60°$.

## 5.2.2. Comparison of BRDF Measurements Results

Based on the method in reference [5], we measured the BRDF of a solar cell and compared the obtained value with the measurement results of the BRDF of a solar panel. The comparison results are shown in Figure 12.

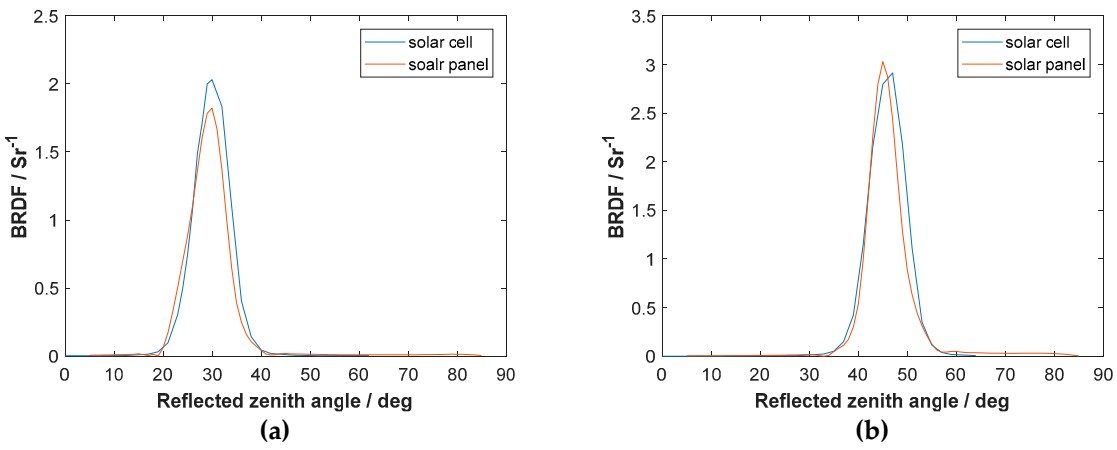

**Figure 12.** *Cont.*

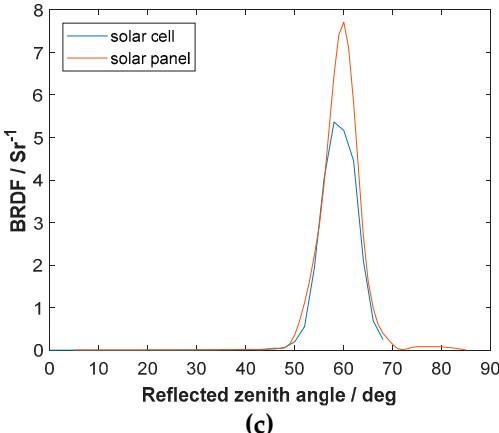

**(c)**

**Figure 12.** Comparison of BRDF measurement results between solar cells and solar panels. (**a**) $\theta_i = 30°$; (**b**) $\theta_i = 45°$; (**c**) $\theta_i = 60°$.

The distance between the two points corresponding to 1/2 of the peak of the BRDF curve was defined as the half width of the BRDF, denoted as $\Delta\theta$. The half-width of the solar cells and solar panels is shown in Table 4.

**Table 4.** Comparison of Half Width of BRDF curve between solar cells and solar panels.

|  | Solar Cells(°) | Solar Panels(°) |
|---|---|---|
| 30° | 8.4108 | 8.1226 |
| 45° | 8.7396 | 6.7717 |
| 60° | 8.8016 | 7.3001 |

As shown in Figure 12, when the incident angle is small (30 degrees and 45 degrees), the difference between the two is small; when the incident angle is large (60 degrees), the difference between the two is obvious. Meanwhile, from Table 4, it can be seen that the half width of the solar panel is less than the half width of the solar cell.

It can also be seen that there is a difference in BRDF between the solar panel with complex material surfaces and the solar cell. At the same time, the necessity of BRDF measurement of solar panel is illustrated.

### 5.3. Measurement Error Analysis

The measurement error has a great influence on the measurement results. There are many factors that will cause errors in the measurement process; the main ones in this experiment are mechanical system errors, illuminating system errors, detection system errors and artificial errors. The errors can be expressed as:

$$\varepsilon_E = \sqrt{\varepsilon_{ME}^2 + \varepsilon_{SE}^2 + \varepsilon_{DE}^2 + \varepsilon_{PE}^2} \tag{14}$$

where $\varepsilon_{ME}$ is the mechanical system error, $\varepsilon_{SE}$ is the illuminating system error, $\varepsilon_{DE}$ is the detection system error and $\varepsilon_{PE}$ is the artificial error.

### 5.3.1. The Mechanical System Error

The mechanical system error can be expressed as:

$$\varepsilon_{ME} = \sqrt{\varepsilon_{Mr}^2 + \varepsilon_{Mh}^2 + \varepsilon_{Md}^2} \tag{15}$$

where $\varepsilon_{Mr}$ is the rotation error of the motorized precision rotary stages, the horizontal rotating table used in this measurement is driven by a stepping motor; its resolution is $0.00022°$. $\varepsilon_{Mh}$ is the adjustment

error of the sample stage with the height error is 0.03 mm and horizontal error is less than 0.01 mm. $\varepsilon_{Md}$ is the displacement error of the detector cantilever. Its absolute value is 0.01 mm. In total, the mechanical system error is less than 3.84%.

### 5.3.2. The illumination System Error

The illumination system error can be expressed as:

$$\varepsilon_{SE} = \sqrt{\varepsilon_{Ss}^2 + \varepsilon_{Su}^2 + \varepsilon_{Sp}^2} \tag{16}$$

where $\varepsilon_{Ss}$ is the error from the stability of the illuminating light source which is less than 0.5% when the light source remains constant for half an hour. $\varepsilon_{Su}$ is the irradiation uniformity error, which is about 2%. $\varepsilon_{Sp}$ is the error from stray light; it was controlled to within 1% during the experimental process. Taking all the above factors into account, the illumination system error $\varepsilon_{SE}$ is approximately 2.29%.

### 5.3.3. The Detection System Error

Because of the limitation of the range of the measurement, we chose a high-intensity light illuminometer to measure the incident illumination on the entrance pupil, and a low-intensity light illuminometer to measure the illumination on scattering directions, as different components will affect the accuracy of the measurements. In order to ensure the validity of the analysis of the error, we took the maximum value of the error caused by the measurement.

The detection system error can be expressed as:

$$\varepsilon_{DE} = \sqrt{\varepsilon_{Dn}^2 + \varepsilon_{Da}^2 + \varepsilon_{Ds}^2}, \tag{17}$$

where $\varepsilon_{Dn}$ is the error from the detector itself. The error of low-intensity light illuminometer was a little large, i.e., 4%. $\varepsilon_{Da}$ is angle error of the illumination detection system from scattering direction, 1.4%. $\varepsilon_{Ds}$ is the measurement error of spot diameter, 2%. So the detection system error was 4.69%.

### 5.3.4. The Artificial Error

The artificial error is caused by different operators. According to the actual measurement of several samples, $\varepsilon_{PE}$ is 1.84%.

Above all, the BRDF measurement error using sun simulator was:

$$\varepsilon_E = \sqrt{\varepsilon_{ME}^2 + \varepsilon_{SE}^2 + \varepsilon_{DE}^2 + \varepsilon_{PE}^2} = 6.74\% \tag{18}$$

## 6. Conclusions

For the BRDF measurement of solar panels with complex periodic material surfaces, a measurement platform was constructed, and a spot size optimization method and measurement radius optimization method are proposed, thereby improving measurement accuracy. The measurement results show that the BRDF of the solar panel has certain specular reflection characteristics, but the half width of the BRDF curve differs from the solar cell. According to the analysis data, the measurement error was about 6.74%. The measurement method can be applied to the measurement of BRDF on other complex periodic material surfaces. The method can also be used to measure and characterize the coatings of heat insulation materials and scattering coating. The measurement data has practical reference value in the evaluation of the the improvement of light absorption of the surface functional materials, such as solar panels, photovoltaic panels and so on. Furthermore, in the field of computer image processing, it can also be used to simulate target image scenes.

**Author Contributions:** Q.H. and Z.W. conceived and designed the experiments together; F.T. and Z.W. built the experimental platform; J.S. analyzed the data; Q.H. wrote the paper.

**Funding:** This work was supported by the National Natural Science Foundation of China (grant no. 61705052) and the Fundamental Research Funds for the Central Universities (grant no. HIT.NSRIF.201630).

**Conflicts of Interest:** The authors declare no conflict of interest.

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
