# Peer review of "Measurement of Equivalent BRDF on the Surface of Solar Panel with Periodic Structure"

_coatings, doi:10.3390/coatings9030193_

Round 1

Reviewer 1 Report

The paper by Hou et al. describes a method of measurement of the bidirectional reflectance distribution function of a solar panel having a periodic structure. The work is certainly of value, but there are certain points that should be considered first before publishing it in Coatings journals. Please take the following remarks into the consideration:

1) Since the whole concept is hinged on the problem of evaluation of a solar panel of complex cross-section, it would be worthwhile to include an examplary cross-section of a solar panel with indication to as which materials constitute each layer. Guide the reader how many of such layers one can expect in a solar cell. At present, your description indicates that the underlying support is the issue (Line 40).

2) Did you draw all the enclosed figures or some of them are taken from the literature? If so, please include appropriate reference in the caption. 

3) It would be good if you could produce a dedicated Experimental section after the Introduction and enclose all the technical details there.

4) Description of the axes in Figure 6 are unreadable. Please correct it.

5) I suggest expanding the Conclusions section to suggest what are the other possible applications of your results. Please say in which fields of science you can expect these findings to be useful as well.

Author Response

Response to Reviewer 1 Comments

Manuscript ID: coatings-459631

Title: Measurement of Equivalent BRDF on the Surface of the Solar Panel with Periodic Structure

Authors: Qingyu Hou, Zhile Wang, Jinyu Su and Fanjiao Tan

Point 1: Since the whole concept is hinged on the problem of evaluation of a solar panel of complex cross-section, it would be worthwhile to include an examplary cross-section of a solar panel with indication to as which materials constitute each layer. Guide the reader how many of such layers one can expect in a solar cell. At present, your description indicates that the underlying support is the issue (Line 40).

Response 1: I have added the cross-section of the solar panel (see Fig 3). As follows:

Figure 3. The cross-section of the solar panel[19]

Line 68-69: The cross-section of the solar panel is shown in Fig 3, including cover glass, ethylene-vinyl acetate (EVA), silicon solar cells with electrode and back glass[19].

[19]. T. Zhang, L. Xie, Y. Li, et al. Experimental and Theoretical Research on Bending Behavior of Photovoltaic Panels with a Special Boundary Condition[J]. Energies. 2018, 11:3435.

Point 2: Did you draw all the enclosed figures or some of them are taken from the literature? If so, please include appropriate reference in the caption.

Response 2: The figures taken from the literature have been enclosed the reference (see Fig.1 and Fig.3). As follows:

Figure 1. Incidence and reflected vectors of BRDF model[6]

[6]. W. Zhang, H. Wang, Z. Wang. Measurement of Bidirectional Reflection Distribution Function on Material Surface[J]. Chinese Optics Letters. 2009, 7(1):88-91.

Figure 3. The cross-section of the solar panel[19]

[19]. T. Zhang, L. Xie, Y. Li, et al. Experimental and Theoretical Research on Bending Behavior of Photovoltaic Panels with a Special Boundary Condition[J]. Energies. 2018, 11:3435.

Point 3: It would be good if you could produce a dedicated Experimental section after the Introduction and enclose all the technical details there.

Response 3: The Measurement method of yellow thermal control material, silver thermal control material has been detailedly described in refenrence [5-9]. The original paper strives to illustrate the difference between solar panel with periodic structure and the uniform material mentioned above on the measurement method and data processing of BRDF. The innovative measurement method and principle are provided in the original paper (see Section 3). I enclose the experimental details in the end (see Page 4). These measurement experimental details are the concrete realization process of the measurement principle, I don’t think it is necessary to add this part and I simplistically describe it in the original paper (see Section 4.2).

Point 4: Description of the axes in Figure 6 are unreadable. Please correct it.

Response 4: I have corrected it and now it’s Figure 12. As follows:

(a)

(b)

(c)

Figure 12. Comparison of BRDF measurement results between solar cells and solar panels

Point 5: I suggest expanding the Conclusions section to suggest what are the other possible applications of your results. Please say in which fields of science you can expect these findings to be useful as well.

Response 5: I have expanded the Conclusion in other possible applications and field, as the red words show below:

For the BRDF measurement of solar panels with complex periodic material surfaces, a measurement platform is built up, a spot size optimization method and measurement radius optimization method are proposed, and measurement accuracy is improved. The measurement results show that the BRDF of the solar panel has a certain specular reflection characteristics, but the half width of the BRDF curve differs from the solar cell. After analysis, the measurement error is about 6.74%. The measurement method can be applied to the measurement of BRDF on other complex periodic materials surface. The method can also be used to measure and characterize the coatings of heat insulation material and scattering coating, the measurement data has practical reference value on evaluating the improvement of light absorption of the surface functional material, such as solar panel, photovoltaic panel and so on. In addition, in the field of computer image processing, it can also be used to simulate the target image scene.

Reviewer 2 Report

Few minor corrections (basically references) are needed for the paper entitled "Measurement of Equivalent BRDF on the Surface of the Solar Panel with Periodic Structure" as below

Author need to provide sufficient references in the introduction.

Provide couple of reference in the first para of intro.

Line: 32-33: Provide references for space target materials such as yellow thermal control material, silver thermal control material and solar cells.

Otherwise the paper is ok.

Author Response

Response to Reviewer 2 Comments

Manuscript ID: coatings-459631

Title: Measurement of Equivalent BRDF on the Surface of the Solar Panel with Periodic Structure

Authors: Qingyu Hou, Zhile Wang, Jinyu Su and Fanjiao Tan

Point 1: Provide couple of reference in the first para of intro.

Response 1: I have provided reference in the first para of intro (see Para 1). As follows:

The bidirectional reflectance distribution function (BRDF) is gradually introduced to describe the reflection characteristics of the material surfaces and is widely used in the target scattering characteristics description[1], radiation calibration[2] and remote sensing imaging[3]. Measurement of optical BRDF on the surface of the space target has great value in the modeling of space target scattering characteristics[4], spectral feature extraction[5] and target identification and tracking.

[1]. B. Zhang, W. Liu, Q. Wei, et al. Analysis of scattering characteristic of the sample based on BRDF experiment measurements[J]. OPTICAL TECHNIQUE. 2006, 32(2):180-182.

[2]. Q. Xue, S. Wang, X. Yang, et al. Spectral radiance responsivities calibration of limb imaging spectrometer[J]. Journal of Optoelectronics Laser. 2010, 21(3):406-410.

[3]. T. Yu, W. Wei, Y. Zhang, et al. Analysis of the BRDF Characteristics of Dunhuang Radiometric Calibration Site in the Spring[J]. ACTA PHOTONICA SINICA. 2018, 47(6):0612004.

[4]. A. Wang, H. Zhang, Z. Wu, et al. Experiment measurements and optimal modeling of goal surface's visible  spectrum BRDF[J]. OPTICAL TECHNIQUE. 2008, 34(5):655-658.

[5]. Y. Yuan, C. Sun, X. Zhang. Measuring and Modeling the Spectral Bidirectional Reflection Distribution Function of Space Target's Surface Material[J]. ACTA PHYSICA SINICA. 2010, 59(3):2097-2103.

Point 2: Provide references for space target materials such as yellow thermal control material, silver thermal control material and solar cells.

Response 2: I have provided reference (see Line: 32-33). As follows:

…then carried out the BRDF measurement of space target materials, such as yellow thermal control material, silver thermal control material and solar cells[10].

[10]. F. Wang, W. Zhang, H. Wang. Reflection Characteristics of On-orbit Satellite Based on Bidirectional Reflectance Distribution Function[J]. Opto-Electronic Engineering. 2011, 38(9):6-12.

Round 2

Reviewer 1 Report

Thank you for including my suggestions. I recommend publication of this article.

Reviewer 2 Report

The paper is in good shape.